# Impact of Nordihydroguaiaretic Acid on Proliferation, Energy Metabolism, and Chemosensitization in Non-Small-Cell Lung Cancer (NSCLC) Cell Lines

**DOI:** 10.3390/ijms252111601

**Published:** 2024-10-29

**Authors:** Carina Chipón, Paula Riffo, Loreto Ojeda, Mónica Salas, Rafael A. Burgos, Pamela Ehrenfeld, Rodrigo López-Muñoz, Angara Zambrano

**Affiliations:** 1Instituto de Bioquímica y Microbiología, Facultad de Ciencias, Universidad Austral de Chile, Valdivia 5090000, Chile; cchipon21@outlook.com (C.C.); paula.riffo@alumnos.uach.cl (P.R.); ojedatampeloreto@gmail.com (L.O.); monicasalas@uach.cl (M.S.); 2Instituto de Farmacología y Morfofisiología, Facultad de Ciencias Veterinarias, Universidad Austral de Chile, Valdivia 5090000, Chile; rburgos1@uach.cl; 3Center for Interdisciplinary Studies on the Nervous System (CISNe), Universidad Austral de Chile, Valdivia 5090000, Chile; ingridehrenfeld@uach.cl; 4Instituto de Anatomía, Histología y Patología, Facultad de Medicina, Universidad Austral de Chile, Valdivia 5090000, Chile

**Keywords:** NDGA, proliferation, cell death, chemotherapy, drug synergism

## Abstract

Lung cancer (LC) is the leading cause of cancer death worldwide. LC can be classified into small-cell lung cancer (SCLC) and non-small-cell lung cancer (NSCLC), with the last subtype accounting for approximately 85% of all diagnosed lung cancer cases. Despite the existence of different types of treatment for this disease, the development of resistance to therapies and tumor recurrence in patients have maintained the need to find new therapeutic options to combat this pathology, where natural products stand out as an attractive source for this search. Nordihydroguaiaretic acid (NDGA) is the main metabolite extracted from the Larrea tridentata plant and has been shown to have different biological activities, including anticancer activity. In this study, H1975, H1299, and A549 cell lines were treated with NDGA, and its effect on cell viability, proliferation, and metabolism was evaluated using a resazurin reduction assay, incorporation of BrdU, and ki-67 gene expression and glucose uptake measurement, respectively. In addition, the combination of NDGA with clinical chemotherapeutics was investigated using an MTT assay and Combenefit software (version 2.02). The results showed that NDGA decreases the viability and proliferation of NSCLC cells and differentially modulates the expression of genes associated with different metabolic pathways. For example, the LDH gene expression decreased in all cell lines analyzed. However, GLUT3 gene expression increased after 24 h of treatment. The expression of the HIF-1 gene decreased early in the H1299 and A549 cell lines. In addition, the combination of NDGA with three chemotherapeutics (carboplatin, gemcitabine, and taxol) shows a synergic pattern in the decrease of cell viability on the H1299 cell line. In summary, this research provides new evidence about the role of NDGA in lung cancer. Interestingly, using NDGA to enhance the anticancer activity of antitumoral drugs could be an improved therapeutic resource against lung cancer.

## 1. Introduction

According to the Global Cancer Observatory (GLOBOCAN), in 2022, lung cancer (LC) was the cancer with the highest rates of incidence and deaths. In 2022, 1.8 million people died from lung cancer, which is twice the number of deaths caused by the second most common cancer in the world, colorectal cancer.

LC is classified into two main histological groups: small-cell lung cancer (SCLC) and non-small-cell lung cancer (NSCLC), accounting for approximately 85% of all lung cancer cases. NSCLC tumors have four stages: I, II, III, and IV. Most cases of LC are diagnosed in the advanced stages (III–IV), which are correlated with poor survival, with a five-year survival of approximately 6% [1]. Besides the significant advances in traditional and molecular therapies, the mortality of LC remains high, which has increased the search for new therapeutic options to treat this disease.

Natural products have always represented an attractive source of new potential drugs. Approximately 60% of anticancer agents come from natural sources [2]. For lung cancer, many natural compounds have antitumor effects, including anti-proliferative, anti-angiogenesis, and anti-metastasis properties, among others. Curcumin, resveratrol, and epigallocatechin-3-galate (EGCG) are some of the most studied natural compounds for their anticancer effects on lung cancer and have the ability to enhance the chemotherapeutic agent’s activity in LC cancer lines [3]. There is also much research on the role of flavonoids in lung cancer, which have antitumoral effects through a wide range of molecular mechanisms, such as cyclin-dependent kinase 1 (CDK1), proliferating cell nuclear antigen (PCNA), and nuclear transcription factor kappa B (NFκB), or inhibiting PI3K/Akt/mTOR, ERK, and Wnt/β-catenin. For example, chalcones induce apoptosis via regulating the expression of HIF-1α, CDK, and MMP-9, and also through downregulating the Akt and extracellular signal-regulated kinase (ERK)1/2 signaling pathways and B-cell lymphoma-2 (Bcl-2) expression [4,5].

NSCLC is a heterogeneous disease, and like other cancers, its treatment has evolved from the use of chemotherapy to the implementation of targeted immunotherapy. Mutations on the EGFR, ALK, BRAF, MET, ROS1, and RET genes are important for the tumorigenesis of NSCLC [6]. However, the primary driver mutations of NSCLC are epidermal growth factor receptor (EGFR), with 10–35% frequency, and Kirsten rat sarcoma (KRAS), with 25% frequency [7]. Several EGFR tyrosine kinase inhibitors (TKIs) (e.g., gefitinib, erlotinib, afatibin, and simertinib) have been developed that have turned out to be more effective than platinum-based chemotherapy in patients with EGFR mutations [8]. In addition, a few KRAS inhibitors have been developed for the treatment of NSCLC, such as sotorasib, the first KRAS G12C inhibitor approved by the FDA in 2021 [9]. Moreover, the development of immune checkpoint inhibitors (ICIs) has revolutionized the treatment of NSCLC patients with the use of monoclonal antibodies (mAbs) such as nivolumab, pembrolizumab (targeting PD-1/PD-L1), and ipilimumab (targeting CTLA-4) [10]. Unfortunately, most patients develop resistance to these target treatments, which has led to the use of different treatment strategies that include the combination of different targeted therapies with chemotherapy, e.g., pembrolizumab plus trametinib, pembrolizumab plus platinum-based chemotherapy, nivolumab plus ipilimumab, and platinum-doublet chemotherapy, among others [11]. However, the overall survival of NSCLC patients remains low. In addition, it should be noted that most of the current clinical trials for NSCLC involve the use of combinations of high-cost drugs, which is why the search for and possible use of natural products in combination with clinical drugs could allow greater accessibility to this type of treatment.

Nordihydroguaiaretic acid (NDGA) is the primary metabolite of *Larrea tridentata*, a bush found in desert areas in Mexico and the United States, and corresponds to approximately 10% of the leaves’ dry weight. The leaves of this shrub have been widely used for infusions in herbal medicine to treat different diseases, including kidney and gallbladder stones, rheumatism, bronchitis, and diabetes, among others [12,13]. Besides its anti-inflammatory and antioxidant effects, NDGA has significant antitumoral effects on several human cancer cell lines, including breast, pancreatic, cervical, prostate, and lung cancer [14]. However, the molecular targets of this polyphenol and the mechanisms by which it exerts its anticancer effect in lung cancer remain unclear. It is important to highlight that in cancer, the cell needs to improve nutrient uptake from the medium to satisfy the higher demands of metabolites necessary for increased proliferation. It has been determined that NDGA inhibits GLUT1 transport in leukemic cells [15], inducing an important impact on cell proliferation. However, the role of NDGA on cell metabolisms in lung cancer must still be elucidated.

Extensive research has suggested that NDGA might mediate anti-cancer effects in different human cancer types through different mechanisms. In human breast cancer cells, NDGA can reduce growth and induce apoptosis partially due to the inhibition of IGF-1R and HER2 at concentrations of 10 mM [16,17]. In cancer prostate cells, NDGA can decrease growth in part by inhibiting IGF-1R at concentrations between 10–30 mM and impair cell motility through suppressing NRP1 [18,19]. In neuroblastoma cells, treatment with NDGA inhibits IGF-1R and induces apoptosis [20]. In pancreatic and cervical human cancer cell lines, NDGA also induces apoptosis, although the mechanism remains unclear; nevertheless, it has been described that inhibition of β-catenin/TCF signaling, TGF-β, and cyclin D may be responsible for this response [21,22,23].

In this work, our principal aim was to investigate the possible mechanisms, specifically the induction of cell death, arrest of the cell cycle, and modulation of glycolytic metabolism, by which NDGA impacts lung cancer cell viability, growth, and metabolism and explore its potential in combinatory treatment with clinical drugs for lung cancer treatment. Analyzing the molecular mechanism associated with the antitumoral effect induced by NDGA on lung cancer in vitro will provide important evidence to fight cancer. At the same time, focusing on analyzing the effect of this molecule alone or in combination will provide a new viable strategy against cancer.

## 2. Results

### 2.1. NDGA Decreases Cell Survival of Non-Small-Cell Lung Cancer (NSCLC) Cells

To evaluate the antitumor role of NDGA, the H1975, H1299, A549, H358, Calu-1, SK-LU-1, and H2228 lung cancer cell lines (NSCLC) were treated with increasing concentrations of NDGA for 72 h. As shown in Figure 1, the NDGA exposure decreased the cell survival of the seven cell lines, where H1975, H358, and Calu-1 cells were the most sensitive, and A549, SKLU-1, and H2228 cells were the most resistant cell lines to NDGA treatment, with a calculated IC50 value between 15–25 and 30–45 µM, respectively. Interestingly, NDGA concentrations over 100 µM reduce cell viability by almost 100% in all cell lines. 

NSCLC can be classified into different subtypes: adenocarcinoma (LUAD), squamous-cell carcinoma (LUSC), or large-cell carcinoma (LCC). For further experiments, we selected the H1975 cell line, the most sensitive LUAD cell line; H1299, the fourth most sensitive LCC cell line; and A549, the most resistant LUAD cell line. 

### 2.2. High Concentration of NDGA Induces Cell Death in NSCLC Cells

To corroborate the cell death process induced by NDGA, propidium iodide staining was performed with different concentrations of NDGA for 24 h in H1975, H1299, and A549 NSCLC cells (Figure 2A). A decrease in cell viability after NDGA treatment was observed, suggesting that treatment with this compound can induce a type of cell death involving the permeabilization of the plasmatic membrane after 24 h of treatment. Table 1 shows the IC50 values for each cell line. 

Since apoptosis is a non-inflammatory death mechanism and typically causes minimal side effects in the body [24], it was first evaluated if treatment with NDGA induces this type of programmed cell death in NSCLC cells. The cell lines were treated with different concentrations of NDGA (IC50 of each cell line, 100 µM, and 250 µM) for 24 h to assess whether NDGA can induce this type of death in NSCLC cells. As displayed in Figure 2B–D, NDGA treatment significantly increases the population of annexin V+/propidium iodide+, which represents late apoptotic cells, in the NSCLC lines assayed. In addition, the treatment increases the population of annexin V–PI+, which represents necrotic cells. In addition to flow cytometry assays, caspase-3 activity was evaluated after 24 h of treatment with IC50 and 100 µM of NDGA (Figure 2E–G). These results show that NDGA treatment did not alter caspase-3 activity in either of the concentrations assayed in the NSCLC cells compared to control cells.

On the other hand, the gene expression of Bcl2 and Bcl-2-associated X protein (Bax), proteins associated with apoptotic cell death, was measured. As observed in Figure 2H–J, the Bcl2/Bax ratio in H1975 and A549 cells decreased after 24 h of treatment with NDGA; however, H1299 cell lines showed an opposite response. All these results suggest that NDGA could induce a non-apoptotic cell death mechanism independent of caspase-3 activation in the NSCLC cell lines. 

Since NDGA seems to induce non-apoptotic cell death, some hallmarks associated with pyroptosis were analyzed because the rupture of the plasma membrane characterizes this cell death type, and the cell also became permeable to propidium iodide (PI) [25]. The cells were treated with the IC50 of NDGA for 3 h, 6 h, and 24 h to analyze the gene expression of proinflammatory cytokines, IL-1b and IL-18, which are increased in the presence of pyroptosis cell death. An increase in the expression of IL-1b was observed (Figure 3A–C) in H1975 and A549 cell lines after 24 h of treatment. In the case of H1299, there is a non-significant increase after 3 h of treatment, with a reduction after 24 h of treatment. Analyzing the expression of IL-18 (Figure 3D–F), the results showed an increase after 3 h of treatment with NDGA in the H1975 cell line and a slight increase after 24 h of treatment in the A549 cell line. Like IL-1b, in H1299 cell lines, NDGA induces a decrease in the expression of IL-18 at all treatment times analyzed. One of the most important hallmarks of pyroptosis is the proteolytic fragmentation of gasdermin, specifically gasdermin E (GSDME) and gasdermin D (GSDMD). The levels of GSDME and GSDMD were analyzed, along with their fragmentation, in response to NDGA treatment using two different concentrations. Fragmentation of these proteins could not be observed, although changes in protein levels were evident. In Figure 3G, the results show a decrease in GSDME levels in H1299 and A549 cell lines, with no differences in H1975. However, for GSDMD, there is a decrease only in the H1975 cell line. This protein was not observed in the H1299 cell line, and we can see an increase in the levels of GSDMD in A549 after NDGA treatment.

### 2.3. NDGA Represses the Proliferative Capacity of NSCLC Cells 

Besides cell death resistance, cancer cells have unlimited proliferative capacity. To analyze the effect of NDGA on this feature of lung cancer cells, the H1975, H1299, and A549 cells were treated with this compound to evaluate whether they could retain proliferative capacity over time. As shown in Figure 4A, treatment with NDGA for 24 h drastically decreased the colony-forming capacity of H1975, H1299, and A549 cells at concentrations lower than the calculated IC50 value. After treatment with 100 µM, cell proliferation is completely suppressed. To corroborate these results, the cells were treated for 24 h with different concentrations of NDGA (close to the IC50 value for each line and 100 µM), and then BrdU incorporation was measured (Figure 4B–D). In concordance with the colony-forming assays, NDGA significantly decreases BrdU incorporation in all three NSCLC cell lines analyzed. The gene expression of Ki-67 (Figure 4E–G), a classical proliferation marker, was also significantly decreased after 24 h of treatment with the IC50 value of NDGA. These results suggest that NDGA has a pronounced antiproliferative capacity over all three NSCLC lines evaluated. To delineate the mechanism by which NDGA inhibits cell growth, the effects of NDGA on the cell cycle of H1975, H1299, and A549 cells were evaluated (Figure 4H–J). After 24 h, treatment with NDGA produces a significant increase in cells in the G1 phase in the H1975 cell line (the most sensitive to NDGA) and a decrease in the population of cells in the G2 phase of the cell cycle in H1975, H1299, and A549 cell lines. 

### 2.4. NDGA Treatment Differentially Modulates Glucose Uptake and Metabolic Enzymes in NSCLC Lines

Along with cell death resistance and unlimited proliferative capacity, reprogramming different metabolic pathways is a classic characteristic of cancer cells. Since it has been described that NDGA causes glucose uptake inhibition in human leukemia cell lines [15], and to investigate the possible effects of NDGA on the metabolism of NSCLC cells, the cell lines were treated for 6 h and 24 h with different concentrations of NDGA before glucose uptake was measured. As displayed in Figure 5A–C, treatment with 100 µM of NDGA decreased glucose uptake in NSCLC cells. According to the results, treatment with NDGA in H1975 cells, the most sensitive cell line, decreased glucose uptake at 6 h of treatment with 25 µM of NDGA. 

To determine the role of NDGA on glucose metabolism, the gene expression of some glycolytic enzymes and transcription factors essential in the metabolic switch in cancer was analyzed. Specifically, H1975, H1299, and A549 cell lines were treated for 3, 6, and 24 h with the IC50 of NDGA, and the mRNA expression of GLUT1, GLUT3, PKM2, HIF-1, and LDH was evaluated by qPCR. As shown in Figure 5D–F, the expression of GLUT1 in H1975 did not change in response to NDGA treatment. Meanwhile, in H1299 and A549, the expression of GLUT1 decreased and increased significantly after 24 h of treatment, respectively. Interestingly, the expression of GLUT3 increased in all cell lines after 24 h of treatment with NDGA (Figure 5G–I). On the other hand, PKM2, the preferent pyruvate kinase isoform expressed in cancer cells [26], decreased significantly in H1975 cells and increased in H1299 and A549 cells after 24 h of treatment with NDGA (Figure 5J–L). The mRNA levels of HIF-1 and LDH, proteins that participate in reprogramming cancer cell metabolism, were also evaluated after treatment with NDGA. As shown in Figure 5M–O, HIF-1 levels decreased in all cell lines after 3 h and 6 h of treatment, and a similar response occurs with the LDH levels (Figure 5P–R). This result suggests that treatment with NDGA impacts the metabolism of the three NSCLC lines assayed. 

### 2.5. NDGA Synergically Enhances the Carboplatin Effect over NSCLC Cells

Some advantages of the use of natural products are that they generally are multi-target drugs and have less toxicity [27], which has led to the study of the combination of these compounds together with chemotherapeutics that are currently used for cancer treatment. Given that there is evidence showing a possible synergistic interaction of NDGA with paclitaxel (also known as taxol), the potential of NDGA to act as an activity enhancer of three chemotherapeutics, carboplatin, gemcitabine, and taxol, was evaluated for lung cancer treatment. To analyze this, H1975, H1299, and A549 cells were treated with NDGA in combination with the chemotherapeutic drugs for 72 h, and cell viability was evaluated by MTT assays, and the results were analyzed in Combenefit software [28,29]. As shown in Figure 6A–C, the combination of NDGA and carboplatin shows a synergic interaction pattern in H1975 and H1299 cell lines, the combination of NDGA and gemcitabine shows synergic interaction only in H1299 cells, and the treatment of NDGA plus taxol shows synergic interaction in H1299 and A549 cells, based on the global score SUM_SYN_ANT_WEIGHTED (considering both antagonistic and synergistic effects) that is calculated using Combenefit software. Interestingly, the combination of NDGA with these three chemotherapeutics drugs shows synergistic interactions in the H1299 cell line between low concentrations of the chemotherapeutics and median concentrations of NDGA. Moreover, the combination with the highest synergy score corresponds to the combination of NDGA and carboplatin in the H1975 cell line, where the synergy pattern occurs between low concentrations of NDGA and carboplatin.

## 3. Discussion

It has been described that NDGA has several biological activities, among which are anti-inflammatory, antioxidant, antibacterial, antiviral, and anticancer properties [13]. In recent years, diverse studies have shown the capacity of NDGA to decrease cell viability and inhibit cell proliferation in several human cancer types, including lung cancer. However, the underlying mechanism and the primary target of this lignan in lung cancer remain unclear. In the current study, we evaluate the effects of NDGA on non-small-cell lung cancer (NSCLC) cells, analyzing changes in cell viability, proliferation, and metabolism. In addition, we determined the synergism of NDGA after combining it with chemotherapeutics used in clinics to treat lung cancer.

In lung cancer, it has been described that NDGA can reduce proliferation and cause apoptosis in both in vitro and in vivo models, in part due to the inhibition of lipoxygenase activity [30]. Inconsistent with this evidence, our results show that NDGA decreased cell viability, with a calculated IC50 value between 25–50 mM, and strongly suppressed H1975, H1299, and A549 cell growth at concentrations even lower than the IC50. 

However, we found that NDGA induces a specific cell death process that involves cell membrane permeabilization, which does not align with apoptosis since treatment with NDGA does not increase the population of annexin V+/PI- cells or caspase 3 activity. These data are related to previous evidence showing that NDGA can inhibit caspase-3 activation in glioma cells, preventing CD95L-induced apoptosis [31]. Many types of cell death involve loss of cell membrane integrity, such as necroptosis, ferroptosis, and pyroptosis [32]. Ferroptosis is triggered by an accumulation of intracellular iron, increasing ROS and lipid peroxides [33]. Since NDGA is an inhibitor of LOX activity, which increases the pool of lipid peroxides, this polyphenol might alleviate and prevent ferroptosis [34,35]. On the other hand, both necroptosis and pyroptosis are types of inflammatory cell death. Pyroptosis and necroptosis are primarily executed by gasdermin and MLKL proteins, respectively. However, necroptosis is a non-apoptotic form of regulated cell death, while pyroptosis can be induced by several caspases, including caspases 1, 3, 7, and 8 [36,37]. Our results show an increase in the expression of some inflammatory cytokines, but NDGA does not induce an increase in the fragmentation of GSDME or GSDMD. Future research is needed to determine if any of these proteins are involved in the death mechanism induced by NDGA. 

Beyond the cytotoxic effect of NDGA, this polyphenol also has antiproliferative activity. For lung cancer cells, there is evidence that NDGA can decrease the proliferation of several SCLC and NSCLC cell lines and even decrease NSCLC tumor growth in vivo [30,38]. Our results corroborate the previous evidence since NDGA could reduce the colony-forming capacity of the three NSCLC cell lines and reduce BrdU incorporation, suggesting that this polyphenol can also stop DNA synthesis. Finally, NDGA treatment also reduces the mRNA levels of the proliferation marker Ki-67. In the H1975 cell line, this effect is partly explained by the arrest of the cell cycle in phase G1, which is caused by the NDGA treatment. This result is not a surprise because NDGA treatment also causes the arrest of the cell cycle in the G1 phase in cervical cancer cells [39], and in fact, derivates of NDGA also cause the arrest of the cell cycle in the G1/S phase in glioma cells [40]. For SCLC cells and breast cancer cells, NDGA has also shown the capacity to arrest the cell cycle in the S phase [41,42]. Although NDGA did not cause a significant increase in the G1/S cell population in H1299 or A549 cells, a possible explanation for the antiproliferative effect on these cell lines can be partially explained by the decrease in GSDME protein levels (Figure 3), since it has been described that the knockdown of GSDME in A549 and H1299 cells (and other NSCLC cell lines) decreased the proliferation of these cells in vitro [43].

Cell metabolism changes constitute a significant characteristic of cancer cells, and the reprogramming of cellular energy plays an important role in how cancer cells respond to therapy, including NSCLC cells [44]. It is well known that natural products can modulate different enzymes of different metabolic pathways such as glycolysis, oxidative phosphorylation, and lipid metabolism, among others [45]. In a cell line model of human leukemia, NDGA can inhibit glucose uptake due to a non-competitive mechanism of GLUT1 inhibition [15]. Here, we explored the role of NDGA on NSCLC cell metabolism and found that a concentration of 100 µM NDGA can diminish glucose uptake in the three cell lines assayed, which represents another possible mechanism for the anti-proliferative effect of NDGA. Since changes in GLUT1 expression were not similar in all cell lines and GLUT3 increases in the three cell lines, possibly as a compensatory mechanism, further investigation is needed to determine how NDGA inhibits glucose uptake in NSCLC lines. On the other hand, the decrease in LDH and HIF-1 mRNA levels in H1975, H1299, and A549 cell lines after treatment with NDGA suggests that this polyphenol can switch the metabolic phenotypes of these cells, like the effect of cetuximab on neck squamous-cell carcinoma cells that inhibit the positive regulation of HIF-1 and LDH-A [46]. Treatment with NDGA could downregulate the expression of PKM2 in H1975 cells and upregulate the expression of H1299 and A549 cells. PKM2 is an enzyme that catalyzes the last step of glycolysis, and its inhibition in NSCLC has antiproliferative effects [47]. The fact that NDGA was able to decrease the mRNA levels of several enzymes that promote glycolytic metabolism in the H1975 NSCLC cell line, the most sensitive one, but not cause the same effect in H1299 and A549 cells, which were more resistant to NDGA treatment, is a phenomenon that needs further investigation to understand how the different characteristics of each NSCLC cell line can present a different response to the same treatment, which mimics the behavior of heterogeneous tumors presented by patients. 

Previous reports have shown that NDGA has additive effects when combined with cisplatin in A549, H460, and SHP77 cells, the first two lines belonging to NSCLC and the last to SCLC. The same study observed that the combination of NDGA with paclitaxel, also known as taxol, showed additive effects on the A549 cell line and synergistic effects on the H460 cell line [42]. Our results show that combining NDGA with carboplatin has antagonistic effects on the A549 cell line but synergistic effects in decreasing the cell viability of H1975 and H1299 cell lines. Moreover, the combination of NDGA with taxol showed synergistic effects mainly in A549 and H1299 cell lines when combining median concentrations of NDGA and taxol. The combination of NDGA with gemcitabine only showed synergistic effects in the H1299 cell line. Combenefit software calculates a synergy score for all the combination drug models (Loewe, Bliss, and HSA). In this work, we use the Loewe synergy score, which corresponds to the difference between the Loewe model-based expected additive effect and the actual effect of the drug combination; if the actual effect of a drug combination is greater than the additive effect, the synergy score is higher than zero [48,49]. Based on this, the combination of NDGA with carboplatin, gemcitabine, and taxol produces synergistic effects in the H1299 cell line, but the best synergy combination corresponds to NDGA plus carboplatin in the H1975 cell line, which has a synergy score of 7.32, since a higher synergy score denotes greater synergy. Previous reports have shown that NDGA has additive effects when combined with cisplatin in A549, H460, and SHP77 cells, with the first two lines belonging to NSCLC and the last to SCLC. The same study observed that the combination of NDGA with paclitaxel, also known as taxol, showed additive effects in the A549 cell line and synergistic effects in the H460 cell line [42]. Regarding the synergic effect of the combination of NDGA with carboplatin, one possible explanation for this phenomenon is that NDGA could act on similar targets of carboplatin and together cause more effective cell death, for example, the action of resveratrol in combination with cisplatin on NSCLC cells [50]. In line with this, it has been documented that PKM2 expression induces resistance to carboplatin treatment in NSCLC cells [51]. As shown in Figure 5, treatment with NDGA reduces the mRNA level of PKM2 in H1975 cells and also significantly increases PDH expression in H1975 cells (non-published data), which belong to the cell line with the highest synergy score for the combination of carboplatin with NDGA, suggesting that the treatment with NDGA could act as a chemosensitizing factor for carboplatin effects.

Since the three NSCLC cell lines present different susceptibility and distinct molecular alterations, we expect that the effects of NDGA on these cell lines are a consequence and depend on a mechanism of action that involves different targets on cells. The H1975 cell line from LUAD harbors mutations on CDKN2A, EGFR, PIK3CA, and TP53, the A549 cell line from LUAD harbors mutations on KRAS (G12S), and WT TP53 and H1299 from LCC have mutations on NRAS and a deletion of TP53. Since the most sensitive line, H1975, has an EGFR mutation, and the other lines do not, it could suggest that NDGA could be sensitizing for mutations such as TKIs [52,53]. In addition, since A549 has KRAS G12S mutations and it has been described that a KRAS mutation is a mechanism of primary resistance to EGFR-TKIs [54], it explains why A549 was more resistant to NDGA treatment. Additionally, it has been described that NDGA inhibits several RTKs, such as IFG-1R in breast and prostate cancer cells [16,19] and FGFR3 in multiple myeloma cells [55]. On the other hand, the fact that the most sensitive cell line, H1975, has a TP53 mutation, the second most sensitive H1299 cell line has null TP53, and the most resistant cell line A549 has WT TP53 suggests that p53 could be a target of NDGA or necessary for the action of NDGA over EGFR, as described for the mechanism of cisplatin [56]. In cervical cancer SiHa cells, NDGA induces cell cycle arrest at the G1 phase, possibly by inducing p21 expression and increasing protein levels of p53 [39].

The promising anti-cancer effects of NDGA over several human cancer cell lines in vitro act in a range from low (1 µM) to high (100 µM) concentrations [39,57,58,59,60], and its antioxidant and antitumoral activity in xenografts and mouse models has been evaluated from 1–100 mg/kg of NDGA [18,61,62]. In addition, the effect of NDGA on non-cancer cells has been shown to be none or minimal [15,63]; results from our lab on the effect of NDGA on HaCaT cells (human keratinocytes) revealed that the IC50 of NDGA was higher than 300 µM. However, adverse effects associated with chronic consumption of high doses of NDGA have also been reported and linked with kidney and liver damage [64]. More recently, a clinical trial of non-metastatic recurrent prostate cancer reports that a dose of 2000 mg/kg of NDGA rarely causes adverse effects, including nausea/vomiting, syncope due to dehydration, and elevated liver function tests in one patient [65]. The present work demonstrated that NDGA represses proliferation at concentrations above 50 µM in the three NSCLC cell lines, and the synergy interactions with chemotherapeutics occur at lower concentrations of NDGA and low concentrations of the clinical drugs. Despite old evidence showing that NDGA slowed NSCLC xenograft growth in nude mice [30], the effect and toxicity of NDGA on NSCLC in vivo models might be re-evaluated.

Given the observed effects of treatment with NDGA on cell viability and proliferation and the evidence that this polyphenol can modulate different proteins associated with signaling pathways that are overactivated in cancer, it would be interesting to continue investigating the key effectors that are regulated by this polyphenol. On the other hand, it is suggested that the effects of the combination of this natural product could have with drugs used in targeted therapies be tested. 

In summary, this research provides new evidence about the role of NDGA in lung cancer. NDGA induces cytotoxicity in different lung cancer cell lines, where metabolic switch could play a role. In addition, it can enhance the anticancer activity of antitumoral drugs, which could be an improved therapeutic resource against lung cancer. Further experiments should be conducted to determine whether NDGA can inhibit tumor proliferation in in vivo models.

## 4. Materials and Methods

### 4.1. Cell Culture 

The cell lines H1975 (ATCC: CRL-5908TM), H1299 (ATCC: CRL-5803™), A549 (ATCC: CCL-185™), H358 (ATCC: CRL-5807™), SKLU-1 (ATCC: HTB-57™), Calu-1 (ATCC: HTB-54™), and H2228 (ATCC: CRL-5935™) were maintained in cell culture bottles at 37 °C with 5% CO_2_ in RPMI-1640 medium supplemented with 10% FBS and 1% penicillin/streptomycin. 

### 4.2. Cell Viability Assays by Resazurin Reduction 

NDGA cytotoxicity was determined using the resazurin reduction assay. A total of 1500 cells per well (H1975, H1299, A549, H358, SKLU-1, Calu-1, and H2228) were seeded in 384-well plates, using the OT-2 laboratory robot. Cells were treated with different concentrations of NDGA in cell culture media (0.5 µM, 0.75 µM, 1.1 µM, 1.7 µM, 2.6 µM, 4 µM, 6.2 µM, 9.4 µM, 15 µM, 22 µM, 33 µM, 50 µM, 74 µM, and 114 µM) and incubated for 72 h. Subsequently, four hours before the end of the treatment time, a 0.1 mM resazurin solution was added and incubated at 37 °C and 5% CO_2_. Fluorescence was measured in the Infinite PRO 200 plate reader (TECAN) with 560 nm as the excitation wavelength and 590 as the emission wavelength. The IC50 values were calculated using the Graphpad Prism 8.0 program. 

### 4.3. MTT Assays and In Vitro Combinations

H1975, H1299, and A549 cells were seeded in 96-well plates (1500 cells/wells). The next day, the cells were treated with different concentrations of NDGA in combination with different concentrations of carboplatin, gemcitabine, or taxol. Experiments were performed using 8 concentrations of each drug (diluted in base two) and all possible combinations between them. After 72 h of treatment, cell viability was evaluated by MTT assays. Four hours before the end of treatment, a solution of 0.5 mg/mL of MTT was added to the cells. Finally, the absorbance was measured at 570 nm with a reference wavelength of 690 nm. The results of cell viability were analyzed in Combenefit software that performs comparisons of drug combinations using the three most common drug combination models: the Loewe additivity model, the Bliss independence model, and the HSA model [66]. The concentration ranges for each drug were NDGA 6.2–114 µM, carboplatin 0.4–50 µM, gemcitabine 1.6–200 nM, and taxol 0.4–50 nM (H1975 cells) and 3.9–500 nM (H1299 and A549 cells). 

### 4.4. Cell Viability Assay with Propidium Iodide 

Viability assays with propidium iodide (PI) were performed in 96-well plates. A total of 5000 cells (H1975, H1299, and A549) were seeded per well in a volume of 100 µL of RPMI-1640 medium without phenol red, supplemented with 10% FBS and 1% Pen/Stp. The cells were treated with different concentrations of NDGA (5 µM, 10 µM, 25 µM, 50 µM, 100 µM, 150 µM, 250 µM, and 500 µM) dissolved in DMSO. As a negative control, DMSO (vehicle) was used, and 100 µL of DMF (Dimethylformamide) was used as a positive or death control. The effects on cell viability were evaluated at 48 h. Once the treatment times with NDGA were completed, 100 µL of HBSS-Ca2+/PI buffer was added to the wells, corresponding to the treatments and negative control, and 2 µL of propidium iodide 500 µM to the wells corresponding to the death control, thus achieving a final equal concentration of propidium iodide in all wells. After incubation at 37 °C for 15 min, the incorporation of propidium iodide into cells was measured by fluorescence determination with wavelength emission at 530 nm and ʎ excitation at 620 nm on the Synergy2 multi-reader (BioTek, Winooski, VT, USA). The percentage of cell viability was calculated with respect to the negative control cells, and they were plotted considering the average of 3 different experiments. The IC50 values were calculated using the Graphpad Prism 8.0 program. 

### 4.5. Flow Cytometry with Double Stain AnnexinV/PI 

For annexin V and propidium iodide (PI) staining, the cells were harvested, washed twice with cold PBS, and stained with FITC Annexin V Apoptosis Detection Kit I according to the manufacturer’s instructions. Briefly, the cells were double stained with annexin V and PI in binding buffer for 15 min at room temperature. The cells were then analyzed using a FACSCanto II flow cytometer (Becton Dickinson, Franklin Lakes, NJ USA). Data were acquired and analyzed with FlowJo 7.6 Software (TreeStar Inc., Ashland, OR, USA). 

### 4.6. Cell Cycle Analysis Using Flow Cytometry 

H1975, H1299, and A549 cells were seeded (1 × 10^6^ cells) in 6-well plates and treated the next day with NDGA (IC50 value and 100 µM) for 24 h. After treatment, the cells were trypsinized, resuspended in PBS, and centrifugated at 600× *g* for 5 min at room temperature. Subsequently, the cells were fixed in 1 mL of cold 3.7% paraformaldehyde + 0.03% Tween-20 for 1 h. After fixation, the cells were washed twice with PBS and then resuspended and stained with 400 µL of PI solution (50 µg/mL) and 50 µL of RNase A solution (100 µg/mL) for 15 min at room temperature in darkness. The cells were then detected using a FACSMelody (Becton Dickinson, Franklin Lakes, NJ, USA) flow cytometer. Cell cycle phases were modeled using the Watson pragmatic algorithm of the cell cycle tool of the FlowJo v10 software (FlowJo LLC).

### 4.7. Caspase-3 Activity Assay 

H1975, H1299, and A549 cells were seeded in 6-well plates (500.000 cells/well) and then treated with different concentrations of NDGA (IC50 and 100 µM) for 24 h. At the end of treatment, the cells were trypsinized and recollected by centrifugation at 600× *g* for 8 min at 4 °C. The cell pellet was washed once with cold PBS, centrifugated at 600× *g* for 8 min, and resuspended in 1× lysis buffer. After 20 min of incubation on ice, the cells were centrifugated at 12,000× *g* for 15 min at 4 °C. The supernatant was recovered to determine caspase-3 activity using the colorimetric caspase-3 assay kit (Cat #CASP3C-1KT, Merck, St. Louis, MO, USA). After overnight incubation at 37 °C, the absorbance was read at 405 nm. Caspase-3 activity in µmol of p-nitro aniline (pNA) released per min per mL of cell lysate or positive control was calculated according to the manufacturer’s instructions.

### 4.8. Colony-Forming Assay

H1975, H1299, and A549 cell lines were seeded in a 12-well plate (500 cells/well) and treated with different concentrations of NDGA for 24 h. After the end of treatment, the media was replaced with fresh medium every 3 days for 10 days. Subsequently, colonies were fixed in 100% cold methanol for 20 min, washed, and stained with a solution of 0.5% of Cristal Violet (CV) for 5 min. The colonies were observed with a loupe (10× magnification). 

### 4.9. 5-Bromo-2′-deoxyuridine (Brdu) Incorporation Assay 

H1975, H1299, and A549 cells were seeded in a 96-well plate (10.000 cells/well in the case of H1299 and A549 cells and 15.000 cells/well for H1975) and treated with different concentrations of NDGA for 24 h. Five hours before the end of treatment, BrdU solution was added and incubated with the cells at 37 °C with 5% CO_2_. Finally, the cells were fixed and the incorporation of BrdU was evaluated with a BrdU Cell Proliferation Assay Kit (Merck), according to the manufacturer’s instructions. 

### 4.10. RNA Extraction and Real-Time Reverse-Transcription PCR (RT-qPCR) 

Cells were seeded in 60 × 15 mm dishes, treated with different concentrations of NDGA, and total RNA was extracted using Trizol reagent (Life Technologies, Waltham, MA, USA) following the manufacturer’s instructions. Total RNA was subjected to RT-PCR. A total 1 µg of total RNA was used to synthesize complementary DNA with the iScript Reverse Transcription kit (Bio-Rad, Hercules, CA, USA). Quantitative RT-PCR (RT-qPCR) was performed in StepOne Real-Time PCR Systems (Applied Biosystems, Waltham, MA, USA) and analysis was performed as described previously [67]. Expression was normalized to ACTh mRNA expression as a housekeeping gene. Oligonucleotide primers for real-time RT-qPCR are described in Table 2. 

### 4.11. Western Blotting 

H1975, H1299, and A549 cells were seeded in 90 × 20 mm plates (1 × 10^6^ cells per well) and allowed to attach for 24 h. The next day, cells were treated with different NDGA concentrations: IC50 value (25 µM for H1975 and H1299 and 50 µM for A549) and 100 µM. Cells were stimulated for 24 h and then washed twice with PBS. Cell lysates were prepared using a protein extraction buffer (Tris-HCl 50 mM pH 7.4; NaCl 150 mM; EDTA 1 mM; 1% TritonX100) with protease and phosphatase inhibitors. Samples were centrifuged for 15 min at 21,000× *g*. Total protein concentration was determined using the Bradford reagent. Subsequently, 50 µg of total protein was denatured at 95 °C for 10 min and resolved using 12–15% SDS-polyacrylamide gel electrophoresis at 90 V. Proteins were electro-transferred to a nitrocellulose membrane at 380 mA for 1 h 30 min. The membrane was blocked using 5% non-fat milk in Tris-buffered saline containing 0.1% Tween 20 for 2 h at room temperature and washed 3 times with TBST. The membranes were incubated with primary antibodies against gasdermin E, gasdermin D, or vinculin as the loading control, in 5% non-fat milk or 5% BSA in 0.1% TBST, at 4 °C overnight. The membranes were washed with 0.1% TBST and incubated with secondary anti-mouse and anti-rabbit immunoglobulin G (IgG) conjugated to horseradish peroxidase (HRP) in 5% non-fat milk TBST 1×. Following incubation, the membranes were washed with 0.1% TBST. Blots were imaged using a Sygene G:BOX system with chemoluminescent WESTERN SUPERNOVA reagent. Western Blotting quantitation was performed using the ImageJ program software version 5.5. 

### 4.12. Glucose Uptake Assay 

Glucose uptake was measured as described previously. Briefly, after exposing cells seeded in a 96-well plate to different concentrations of NDGA (25, 50, or 100 µM), the cells were washed with PBS and incubated with 1 mM of 2-deoxyglucose (2-DG) for 10 min at room temperature. Subsequently, 25 µL of stop and neutralization buffer was added followed by 100 µL of 2DG6P Detection Reagent. The plate was incubated for 30 min at room temperature. The 2-deoxuglucose was measured in H1975, H1299, and A549 cells using a glucose uptake-GloTM assay kit (Promega, Madison, WI, USA) and luminescence intensity (Relative light unit, RLU, Clermont-Ferrand, France) according to the manufacturer’s instructions. 

### 4.13. Statistical Analysis 

All experiments were repeated 3 times for each assay and the results were represented as the mean ± SD. One-way ANOVA, two-way ANOVA, or *t*-tests were used to evaluate quantitative data for statistical significance. *p* < 0.05 was considered to indicate a statistically significant difference.

## Figures and Tables

**Figure 1 ijms-25-11601-f001:**
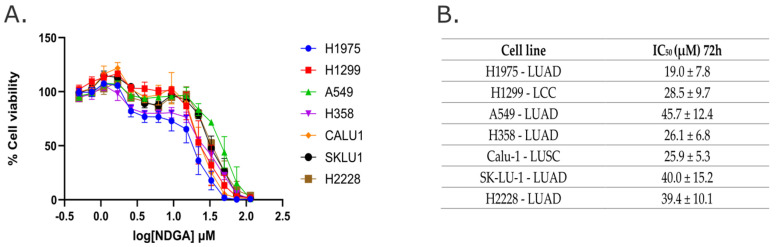
**NDGA decreases cell survival of non-small-cell lung cancer cell lines.** H1975, H1299, A549, H358, Calu-1, SK-LU-1, and H2228 cell lines were treated with different concentrations of NDGA for 72 h (**A**), and the effect on cell viability was measured using a resazurin reduction assay. (**B**) Table with IC50 values obtained from A. Graph data represent mean ± SEM of three independent experiments.

**Figure 2 ijms-25-11601-f002:**
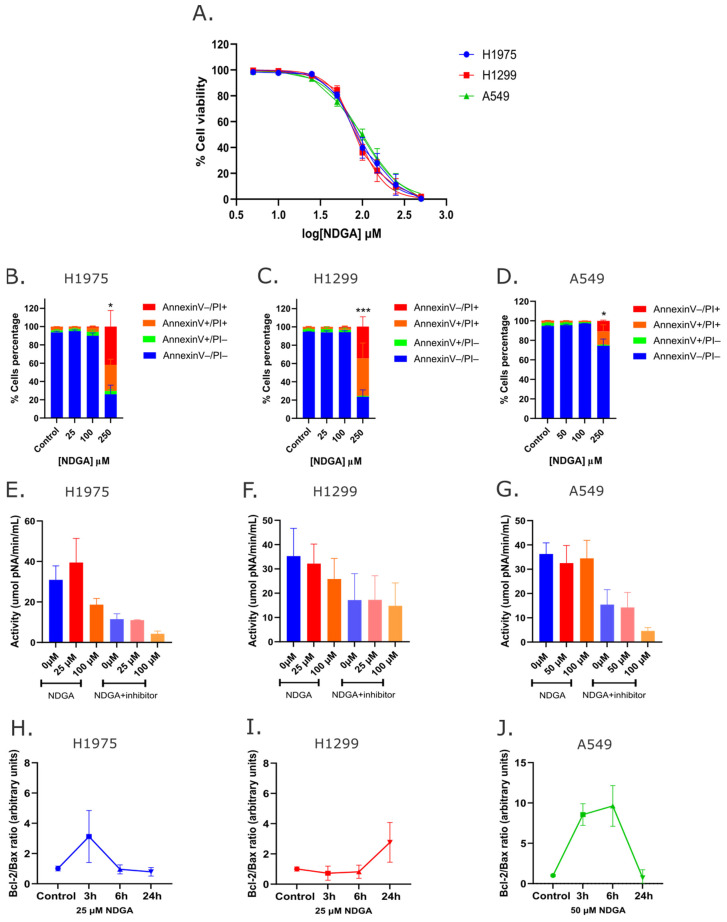
**NDGA induces cell death in NSCLC cells.** (**A**) H1975, H1299, and A549 cells were treated with different concentrations of NDGA for 24 h, and the effect on cell viability was measured by propidium iodide staining. Graphs represent the mean of three experiments performed in triplicate. (**B**–**D**) H1975, H1299, and A549 cells were treated with different concentrations of NDGA (0, IC50, 100, and 250 µM) for 24 h, and the percentage of annexin V+/− or PI+/− cells was evaluated by flow cytometry and graphed. (**E**–**G**) After 24 h of treatment with NDGA, cells were collected by centrifugation, and caspase-3 activity was analyzed according to the kit manufacturer’s instructions. The inhibitor was included in the kit. (**H**–**J**) Cells were treated with the corresponding IC50 values for each cell line for 3 h, 6 h, or 24 h, and then RNA was extracted. The gene expression of Bcl-2 and Bax was evaluated using RT-qPCR assays. Actin was used as a housekeeping gene, and culture media was used as a vehicle. Data are presented as mean ± SEM of 3 independent experiments. Statistical analysis was performed using two-way ANOVA for flow cytometry assays. *p*-values are * *p* < 0.05 and *** *p* < 0.001.

**Figure 3 ijms-25-11601-f003:**
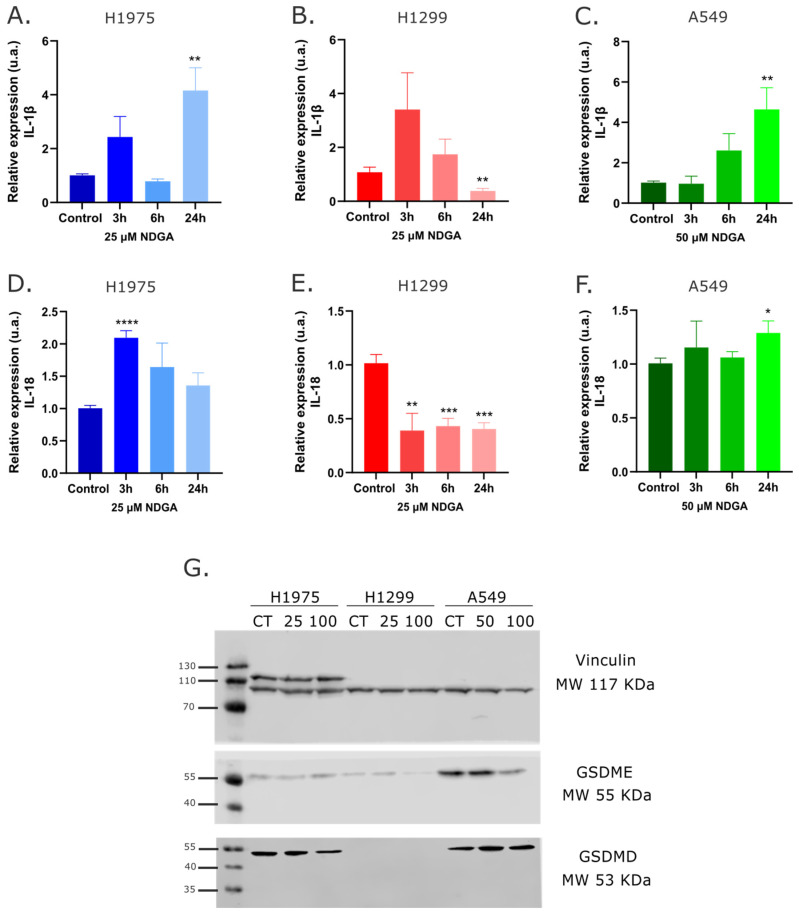
**Role of NDGA on pyroptosis cell death.** (**A**–**C**) Cells were treated with NDGA values corresponding to the IC50 for each cell line during 3 h, 6 h, and 24 h, and then RNA was extracted. The gene expression of IL-1b was evaluated using RT-qPCR assays. (**D**–**F**) Cells were treated with corresponding IC50 values of NDGA for each cell line. The gene expression of IL-18 was evaluated using RT-qPCR assays. Actin was used as a housekeeping gene, and media was used as a vehicle. Data are presented as means ± SD. (**G**). Western blot analysis of total proteins from cells treated with 25 or 100 µM NDGA during 24 h, using the specific antibodies anti-GSDME and anti-GSDMD. An antibody against vinculin was used as a loading control. The cells were treated with a vehicle (media) as a control. Statistical analysis by one-way ANOVA. *p*-values are * *p* < 0.05, ** *p* < 0.01, *** *p* < 0.001, and **** *p* < 0.0001.

**Figure 4 ijms-25-11601-f004:**
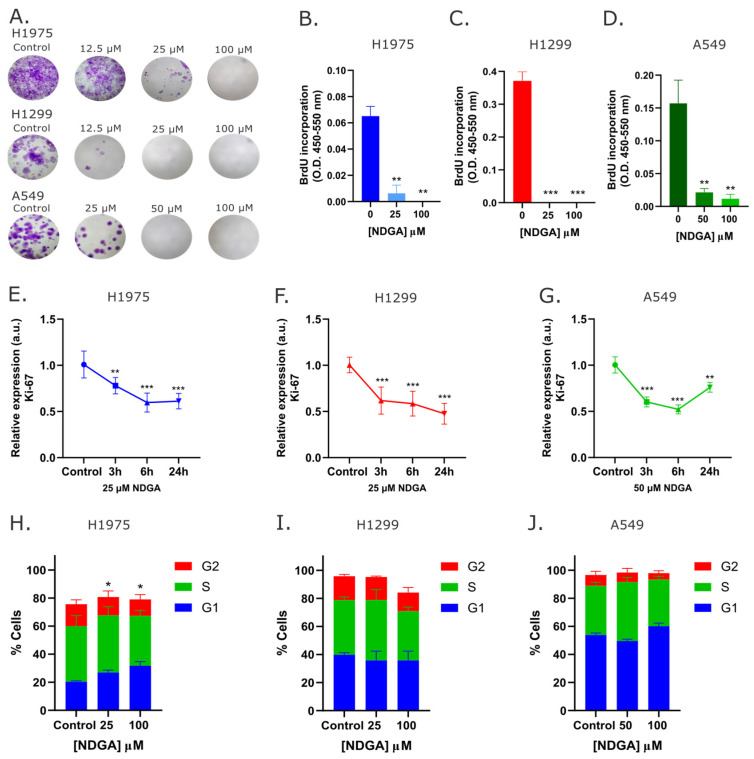
**NDGA inhibits cell proliferation of NSCLC cells.** (**A**) H1975, H1299, and A549 cells were treated with different concentrations of NDGA for 24 h. After 10 days, the colony-forming assay was performed. (**B**–**D**) The cell lines were treated with 25, 50, or 100 µM NDGA for 24 h, and cell proliferation was measured by the relative incorporation of BrdU. (**E**–**G**) Cells were treated for 3 h, 6 h, and 24 h, and then the effect on ki-67 gene expression was evaluated with RT-qPCR assays. Actin was used as a housekeeping gene, and media was used as a vehicle. Data are presented as means ± SD. (**H**–**J**) The cells were treated for 24 h with IC50 and 100 µM NDGA, collected, and stained with PI for cell cycle analysis by flow cytometry. *p*-values are determined by one-way ANOVA compared to the control * *p* < 0.05, ** *p* < 0.01 and *** *p* < 0.001.

**Figure 5 ijms-25-11601-f005:**
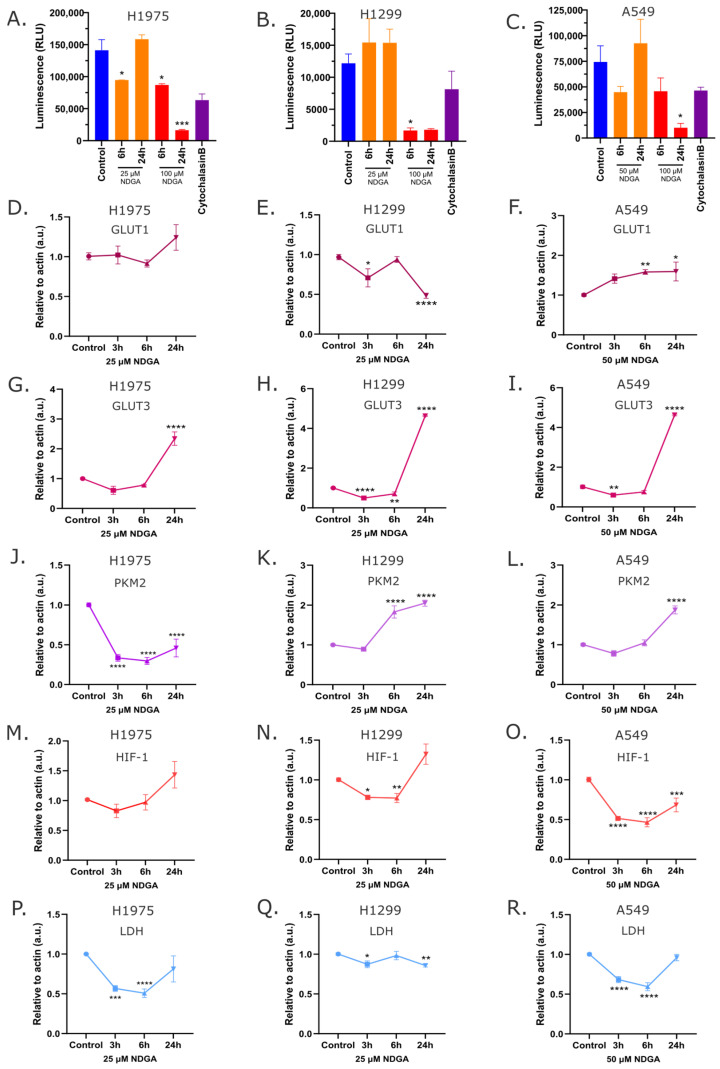
**NDGA impacts glucose uptake and metabolic enzymes.** (**A**–**C**) H1975, H1299, and A549 cells were treated with IC50 values and 100 µM of NDGA for 6 h and 24 h. The intracellular accumulation of 2-deoxy-glucose-6-phosphate was measured by luminescence. Cytochalasin B (50 µM) was used as a glucose transport inhibitor control. (**D**–**R**) H1975, H1299, and A549 cells were treated during 3, 6, and 24 h with IC50 values of NDGA, and then RNA was extracted and used to evaluate the levels of GLUT1 (**D**–**F**), GLUT3 (**G**–**I**), PKM2 (**J**–**L**), HIF-1 (**M**–**O**), and LDH (**P**–**R**). Actin was used as a housekeeping gene, and DMSO (or media) was used as a vehicle. Data are presented as means ± SD. *p*-values were determined by one-way ANOVA compared to the control * *p* < 0.05, ** *p* < 0.01, *** *p* < 0.001, and **** *p* < 0.0001.

**Figure 6 ijms-25-11601-f006:**
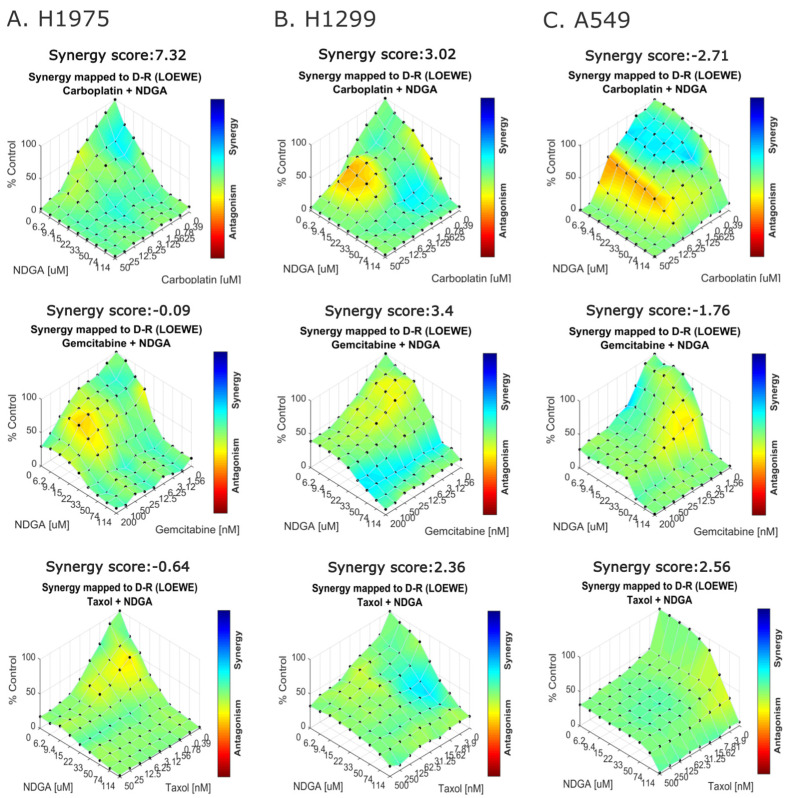
**In combination with carboplatin, NDGA synergically decreases the cell viability of H1975, H1299, and A549 cells.** NSCLC cells were treated for 72 h with different concentrations of NDGA and carboplatin, gemcitabine, or taxol. Cell viability was measured using an MTT assay. (**A**–**C**) Effect matrices are plotted using Combenefit software and the Loewe model of drug combinations.

**Table 1 ijms-25-11601-t001:** IC50 values obtained from PI assays. IC50 table corresponds to the mean of three independent experiments analyzed in triplicate.

Cell Line	IC_50_ (µM) 24 h
H1975	90.8 ± 18
H1299	89.5 ± 15
A549	99.4 ± 19

**Table 2 ijms-25-11601-t002:** Primer sequences.

Primers	Sequence 5′-3′
ACTh Forward	AGATCAAGATCATTGCTCCTC
ACTh Reverse	GGGTGTAACGCAACTAAGTC
BAX Forward	GGCAGCTGACATGTTTTCTGAC
BAX Reverse	CACCCAACCACCCTGGTCTT
BCL2 Forward	TTGAGTTCGGTGGGGTCAT
BCL2 Reverse	GACTTCACTTGTGGCCCAG
GLUT1 Forward	CTTCACTGTCGTGTCGCTGT
GLUT1 Reverse	CCAGGACCCACTTCAAAGAAA
GLUT3 Forward	GCTATGGCCGCTGCTACTGGG
GLUT3 Reverse	CCACAACCGCTGGAGGATCTGC
LDH Forward	ATCTTGACCTACGTGGCTTGGA
LDH Reverse	CCATACAGGCACACTGGAATCTC
HIF1 Forward	CGTTCCTCCGATCAGTTGTC
HIF1 Reverse	TCAGTGGTGGCAGTGGTAGT
KI67 Forward	CGTCCCAGTGGAAGAGTTGT
KI67 Reverse	CGACCCCGCTCCTTTTGATA
PKM2 Forward	ATTATTTGAGGAACTCCGCCGCCT
PKM2 Reverse	ATTCCGGGTCACAGCAATGATGG
IL-1β Forward	TCCTGTCCTGCGTGTTTGAA
IL-1β Reverse	TCTTTGGGTAATTTTTGGGATCT
IL-18 Forward	TGCATCAACTTTGTGGCAAT
IL-18 Reverse	ATAGAGGCCGATTTCCTTGG

## Data Availability

The datasets generated during and/or analyzed during the current study are available by request; please contact the corresponding authors.

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
