# Peer review of "Impact of Nordihydroguaiaretic Acid on Proliferation, Energy Metabolism, and Chemosensitization in Non-Small-Cell Lung Cancer (NSCLC) Cell Lines"

_ijms, 2024, doi:10.3390/ijms252111601_

Round 1

Reviewer 1 Report

Comments and Suggestions for Authors

The manuscript is well wrtiten, interesting, methdologically correct. The results are clear as well as the discussion.

However I have some comments:

- considering the mechanism of action of NDGA, how it can impact the oxidative phosporylation, the mitochondrial metabolism and the production of Reactive oxygen species as inducers of cell death, as yet reported for cisplatin (see PMIF 23876094)

- line 385, the ability to modulate mitochondrial posphorylation and the production of reastive oxygen specie can explain the synergism with cisplatin. Can the authors add these mechanism in the Discussion.

Reviewer 2 Report

Comments and Suggestions for Authors

·         Title: It looks like there are two topics in the same paper. It needs to be clear, specific and reflects the content of the manuscript (e.g., in vitro, in vivo).

·         Line 32-35: Need to provide give more methodological details here.

·         Statistical analysis used should be clearly stated in abstract.

·         Line 36-37: Define NSCLC cells, genes, and metabolic pathways

·         Line 37: How the combination of NDGA with carboplatin shows synergistic interactions. Please clarify.

·         In introduction, please put more focus on the progress during last years on the combination treatments of NSCLC with chemo+ IO and targeted therapies for mutated patients like EGFR/ALK/ROS1/KRAS pts. Please also provide more details on the molecular biology of NSCLC.

·         Line 55-61: It would be interested if that authors state the molecular role of these compounds and other flavonoids (e.g., quercetin, chalcone...etc) with the variety of pathways affected in NSCLC lines (Int J Mol Sci. 2019 Sep; 20(17): 4291; Curr Issues Mol Biol. 2024 Jun 13;46(6):5894-5908).

·         The novelty and/or significance of this study could be improved in the last paragraph of introduction. What does the study add to the literature?

·         Line 472: Please expand on phase cell cycle arrest using this analysis.

·         Please reduce using the term "we" throughout (e.g., we realize, we observed, we first, we could not see...etc.).

·         Please define abbreviations in the first used (e.g.,  Bcl2 and Bax).

·         Line 312-323: This could be moved to the introduction.

·         I miss a discussion on NDGA cytotoxicity (low or high toxic). Please discuss this with other in vivo/vitro studies.

·         The limitations of study were not reported. This should be in a separate section.

·         I would suggest adding a conclusion section with the consequences of the results obtained and recommendations for the future studies.

·         Ref 34: Please add journal name, issue and volume.

·         Ref# 22-24 are very old- please delete.

Reviewer 3 Report

Comments and Suggestions for Authors

The paper, titled "Nordihydroguaiaretic acid (NDGA) prevents cellular survival and proliferation on non-small cell lung cancer (NSCLC) cells. Impact on energy metabolism and chemosensitizing effect," discusses the effects of NDGA on NSCLC cells, focusing on its potential as a treatment. The paper addresses the ongoing challenge of high mortality rates in lung cancer, especially non-small cell lung cancer, due to resistance to existing treatments.

Concerning methodological limitations, the study primarily relies on in vitro cell line experiments, which may not fully replicate the complexity of NSCLC in vivo. Additionally, the study mentions that while NDGA shows potential, the exact molecular mechanisms of its action, particularly its impact on different NSCLC subtypes, require further investigation. Some variability in cell line responses to NDGA was noted, which suggests more detailed mechanistic studies are necessary.

Since the research was funded by specific academic grants and the authors note no conflicts of interest, bias seems minimal. However, a potential bias might exist in overemphasizing NDGA's positive results without equally discussing the variability in cell line responses or the challenges in translating in vitro findings to clinical applications.

Another point of view could be that one could argue that the observed effects on NSCLC cells might be due to non-specific cytotoxicity rather than a targeted mechanism of action. Also, the variability between cell lines suggests that NDGA's effectiveness may depend on specific genetic or metabolic traits of the cells, which are not fully explored in the paper.

Concerning some unresolved questions, the exact molecular pathways through which NDGA exerts its effects on NSCLC cells remain unclear. Future studies should investigate the role of metabolic reprogramming in NDGA's mechanism of action and explore its effects in vivo.

If NDGA's chemosensitizing effects are validated in further studies, it could be developed as an adjunct therapy to improve the efficacy of existing lung cancer treatments, potentially reducing the doses required and minimizing side effects associated with chemotherapy.

Round 2

Reviewer 2 Report

Comments and Suggestions for Authors

No further comments.

Author Response

Comment 1:

No further comments.

Response 1: We are very grateful for the helpful comments in the first round.

Reviewer 3 Report

Comments and Suggestions for Authors

The authors provided a revised version of their paper. Despite some parts have been introduced or re-written, there are still some issues to face, such as the format of figures. Please check figure 1 that is not totally visible.

Moreover, FIgure 2B should be reported as a table and not a figure. Please change it.

Line 360. "in vitro and in vivo" the expressions "in vitro" and "in vivo" should be reported in italics. Same in Line 473.

Author Response

Comment 1: The authors provided a revised version of their paper. Despite some parts have been introduced or re-written, there are still some issues to face, such as the format of figures. Please check figure 1 that is not totally visible.

Response 1: Thanks for the comment. The error was fixed.

Comment 2: Moreover, FIgure 2B should be reported as a table and not a figure. Please change it.

Response 2: Thanks for the comment. We included a table with the IC50 information after figure 2.

Comment 3: Line 360. "in vitro and in vivo" the expressions "in vitro" and "in vivo" should be reported in italics. Same in Line 473.

Response 3: Thanks for the comment. The error was fixed.